# Arsenic-Containing Medicine Treatment Disturbed the Human Intestinal Microbial Flora

**DOI:** 10.3390/toxics11050458

**Published:** 2023-05-15

**Authors:** Jiaojiao Li, Xinshuo Chen, Shixiang Zhao, Jian Chen

**Affiliations:** 1College of Ecology and Environmental Sciences & Yunnan Key Laboratory for Plateau Mountain Ecology and Restoration of Degraded Environments, Yunnan University, Kunming 650500, China; jiaojiaoli@ynu.edu.cn (J.L.); chenxinshuo5223@163.com (X.C.); 2Hematology Department of First People’s Hospital of Yunnan Province, Kunming 650032, China; 3Department of Cellular Biology and Pharmacology, Herbert Wertheim College of Medicine, Florida International University, Miami, FL 33199, USA; 4Institute of Environmental Remediation and Human Health, College of Ecology and Environment, Southwest Forestry University, Kunming 650224, China

**Keywords:** ATO, ATRA, AATO, *Bacteroides fragilis*, enterohepatic circulation, human gut microbiome

## Abstract

Human intestinal microbiome plays vital role in maintaining intestinal homeostasis and interacting with xenobiotics. Few investigations have been conducted to understand the effect of arsenic-containing medicine exposure on gut microbiome. Most animal experiments are onerous in terms of time and resources and not in line with the international effort to reduce animal experiments. We explored the overall microbial flora by 16S rRNA genes analysis in fecal samples from acute promyelocytic leukemia (APL) patients treated with arsenic trioxide (ATO) plus all-trans retinoic acid (ATRA). Gut microbiomes were found to be overwhelmingly dominated by Firmicutes and *Bacteroidetes* after taking medicines containing arsenic in APL patients. The fecal microbiota composition of APL patients after treatment showed lower diversity and uniformity shown by the alpha diversity indices of Chao, Shannon, and Simpson. Gut microbiome operational taxonomic unit (OTU) numbers were associated with arsenic in the feces. We evaluated *Bifidobacterium adolescentis* and *Lactobacillus mucosae* to be a keystone in APL patients after treatment. Bacteroides at phylum or genus taxonomic levels were consistently affected after treatment. In the most common gut bacteria *Bacteroides fragilis*, arsenic resistance genes were significantly induced by arsenic exposure in anaerobic pure culture experiments. Without an animal model, without taking arsenicals passively, the results evidence that arsenic exposure by drug treatment is not only associated with alterations in intestinal microbiome development at the abundance and diversity level, but also induced arsenic biotransformation genes (ABGs) at the function levels which may even extend to arsenic-related health outcomes in APL.

## 1. Introduction

Arsenic is ranked first on the ATSDR 2019 substance priority list (https://www.atsdr.cdc.gov/spl/index.html (accessed on 11 July 2022)). Inorganic arsenic can easily across the cell membrane where it is converted into various arsenic forms. Trivalent arsenic (As(III)) has a higher toxicity than pentavalent arsenic (As(V)), determined by chemical form. [1]. As a consequence of arsenic ubiquity, arsenic biotransformation genes (ABGs) were found largely in bacterial arsenic resistance (*ars*) operons for resistance to inorganic arsenic [2]. The *ars* operon is minimally comprised of *arsRBC* [3], which codes for an ArsR transcriptional repressor, an *arsB* As(III) efflux permeases [4], and an ArsC arsenate reductase, respectively, with the operon negatively regulated ArsR. The genetic repertoire of the arsenic resistance network is predicted to be expanded after The Great Oxidation Event [5,6]. The enlarged *arsRDABC* operon has been found in many bacterial genomes and is well-characterized [7]. *arsD* gene encodes a metallochaperone for delivering As(III) to the ArsAB efflux pump [8]. ArsC reduces As(V) to As(III) which can then be pumped out of the cell for detoxification [9].

Although arsenic can be poisonous, arsenicals have been used in traditional Chinese medicine for thousands of years ago [10]. From 1878 to 1910, Thomas Fowler’s and Salvarsan, an organic arsenical (As_2_O_3_, ATO) were used to treat diseases [11]. In the mid-20th century, arsenic was used to treat acute promyelocytic leukemia (APL) patients who were resistant to all-trans-retinoic acid (ATRA) [12]. APL is a blood cancer in which the development of white blood cells is abnormal to form neutrophil type (promyelocytes) due to a translocation of the promyelocytic leukemia gene (*PML*) on chromosome 15 and the retinoic acid receptor-*α(RARα)* gene on chromosome 17 [13]. The long-term safety of As_2_O_3_ for patients was still not clear, although it was an effective therapy in APL [14]. In 2000, As_2_O_3_ was approved for relapsed or refractory APL by FDA of U.S. [15]. It was reported that ATRA–ATO treatment contributed to higher hepatic toxicity although it has less hematologic toxicity and infections [16]. In the hospital, APL patients were assigned to receive 0.15 mg/kg daily As_2_O_3_ and 45 mg/m^2^/day ATRA by intravenous (iv) until complete remission [16]. Arsenic was eliminated by urine and feces, and the urine/feces ratio changed very substantially due to different routes of administration. Arsenic through iv taking in rat model is metabolized in the liver and excreted into bile by enterohepatic circulation [17].

The human intestinal microbiome, which has been named an “exteriorized organ” [18], harbors ~500–1000 distinct bacterial species. Intestinal microbiome has function not only to maintain intestinal homeostasis but also to affect the metabolism of xenobiotics [19], including toxins, pollutants, drugs, etc. Firmicutes and *Bacteroidetes* were the predominating organisms in the human colon [20], while Bacteroides made up roughly 25% of the entire gut microbiome [21]. Bacteroides have function to keep normal intestinal physiology and function, such as glycan utilization [22], and capturing vitamin B-12 [23]. *Bacteroides fragilis* is isolated from anaerobic infections and is characterized as one of the opportunistic pathogens to produce hemolysins [24]. *B. fragilis* and *B. stercoris* were identified as keystone taxa to influence the gut microbiome structure [25,26]. *B. fragilis* contribute to 70% of Bacteroides infections although it comprises only 2% of the total Bacteroides in the gut [21,27]. *Bacteroides thetaiotaomicron* (formerly *Bacillus thetaiotaomicron*), an anaerobic symbiont found in the human intestine, is considered a keystone taxon based on empirical evidence [28,29]. Confusingly, it has been proposed that the gut microbiome can both mitigate and exacerbate arsenic toxicity [19]. On the one hand, human gut bacteria biochemically metabolize arsenic-containing compounds to mitigate the toxicity of arsenic to the host. It was reported that the microbiome protected host from arsenic-induced mortality in mouse models, and microbiome stability [30] and the presence of specific bacterial such as *Faecalibacterium biofidobactrium* and Lactobacillus were the main reasons for this protection [31]. On the other hand, arsenic toxicity can be exacerbated by the gut microbiome. In vitro studies found that arsenic can be biotransformed by human colonic microbiota into more toxic monomethylarsonous acid (MAs(III)) [32].

Furthermore, research supported that the gut microbiome was disturbed after arsenic exposure in mouse models [33,34]; at the same time, arsenic metabolic profiles were substantially altered by gut microbiome at the function level [35,36]. It was reported that ingested arsenicals interacted with the gut microbiome of mice, causing Firmicutes bacteria to significantly decline while *Bacteroidetes* families remained relatively unchanged [35] or increased [33]. This suggests that Bacteroides species may be intrinsically resistant to arsenic [35]. Most importantly, it was confirmed that Bacteroides had arsenic resistance and the potential ability to methylate arsenic by simulated gastrointestinal tract experiment [37]. Researchers also reported that gut microbiome diversity and arsenic biotransformation function of soil fauna was altered when exposed to arsenic [38]. The previous research also demonstrated that the gut microbiota of earthworms was disturbed by arsenic and provides some insights into how arsenic is biotransformed in the earthworm gut [39]. So far, lots of animal (nonhuman) physiological research provides proof of the adverse effects of arsenic on the gut microbiome. But most animal experiments are onerous in terms of time and resources and not in line with the international effort to reduce animal experiments. Even while studies using mice or a simulated gastrointestinal tract have shown how arsenic alters the gut microbiome [33,37,40], the disturbance from human body exposure is less clear. Due to human ethics limitations, few clinical or epidemiological studies have investigated arsenic exposure effects on human gut microbiome. Furthermore, unlike animal exposure studies in which animals were maintained unified growth conditions and food supply, human bodies are variable in diet and living environment. It is unknown if the effects of arsenic on the human gut microbiome varied depending on inter-individual variance. Despite arsenic effects on the gut, the microbiome has been reported, how the complexities of the human gut bacterial community change while shifting before and after drug treatment in APL states are far from being fully understood. So far, the impact of AATO (ATO+ATRA), a mix of arsenic-containing medicine, treatment on human gut microbiota health is still unknown.

Is arsenic-containing medicine exposure associated with an altered gut microbiome? Thus, we comprehensively define the gut bacterial community structure changes, the dominant microbiota, and the potential functional species after arsenic-containing medicine treatment by sequencing 16S rDNA V4 hypervariable regions from fecal samples of APL patients. We identified that Bacteroides were kept upregulated after taking arsenic-containing medicine, and overexpression of arsenic biotransformation genes (ABGs) *arsA/B/C/D* was demonstrated by pure *B. fragilis* culture. The association between gut microbiome OTU number and levels of arsenic in feces was also assessed. Without taking arsenicals passively, this research provides evidence for the relationships between arsenic exposure and changes in the human microbiota, and even extends to arsenic-related health outcomes in APL.

## 2. Material and Methods

### 2.1. APL Patients

A total of 22 samples were recruited for this study (Table 1). Newly diagnosed APL patients aged 40–60 years (male and female) were included in this study. The APL diagnosis was determined by professional doctors with the following criteria: white-cell count ≤10 × 10^9^ per liter [16,41]; the basis of white-cell morphologic features [16,41]; genetic diagnosis of *PML-RARɑ* fusion gene [42] by qPCR assay [43] or using conventional karyotyping or fluorescence in situ hybridization (FISH) [44], or a microspeckled PML pattern from an indirect immunofluoresence assay [45]. The exclusion criteria were as follows: pregnancy or breastfeeding, suffering from metabolic or gastrointestinal diseases, and history of using antibiotics in recent two months before sampling.

### 2.2. Sample Collection

APL patients were treated with As_2_O_3_ combined retinal acid by iv as follows: 0.15 mg/kg daily As_2_O_3_ and 45 mg/m^2^/day ATRA for induction and consolidation therapy for at least 28 days until complete remission [16]. Participants diagnosed as APL were asked to provide crude urine and fecal samples as long as in the hospital diagnosed as APL (the first-round urine and fecal samples: before taking As_2_O_3_). To examine the effect of ATO exposure on the gut microbiome of ALP patients, group A fecal samples (n = 5) were collected before ATO treatment and group B fecal samples (n = 15), as the second-round urine and fecal samples, were collected within 14th to 21st days after patients had taken arsenic-containing medicine at First People’s Hospital of Yunnan province. Volunteers were requested to provide answers to a screening questionnaire consisting of questions designed to gather basic information on the health of the residents, medical histories, and age and marriage. Samples were immediately frozen in the hospital freezer and stored at −80 °C until analysis. Sample collection for studies was approved by the Ethics Committee of First People’s Hospital of Yunnan Province New Kunhua Hospital. The Manual of Procedures for Human Microbiome Project (http://hmpdacc.org/resources/tools_protocols.php (accessed on 20 March 2020)), with minor modifications, was followed when we collect fecal samples of APL patients. Sample collection for studies was approved by the Ethics Committee of First People’s Hospital of Yunnan Province New Kunhua Hospital. This study strictly followed the guidelines of the Helsinki Declaration and the International Code of Ethics for Biomedical Research Involving Humans, jointly formulated by the World Health Organization and the Council of International Medical Science Organization. All participants provided written informed consent.

### 2.3. Experimental Flow

The Beijing Genomics Institute (BGI) performed the 16S rRNA gene sequencing. DNA from fecal pellets collected from APL patience was amplified using primer 515F (5′-GTGCCAGCMGCCGCGGTAA-3′) and 806R (5′-GGACTACHVGGGTWTCTAAT-3′) to target the V4 regions of 16S rRNA of bacteria. PCR reaction was carried out with 30 ng DNA samples and primers. The PCR amplification products were purified by Agencourt AMPure XP magnetic beads, and labeled, and then library building was completed. The range and concentration of fragments in the library and qualified libraries sequencing based on inserted fragment size were measured by Agilent 2100 Bioanalyzer and HiSeq platform. The 16s sequencing data were deposited in NCBI with Submission ID: SUB12873522, BioProject ID: PRJNA935705 (Raw sequence reads project Accession: PRJNA935705) with link https://www.ncbi.nlm.nih.gov/bioproject/?term=PRJNA935705 (accessed on 27 February 2023).

### 2.4. Statistical Analysis of Data

In the Wilcoxon rank-sum test by R (version 3.2.1), alpha diversity indices, including Shannon, Simpson, Chao, and ACE, were calculated at 97% identity [46]. A principal component analysis (PCA) was also conducted to evaluate the similarity of bacterial communities [47]. MEGAN (version 4.70.4) and GraPhlAn software (https://huttenhower.sph.harvard.edu/graphlan (accessed on 20 July 2021)) were used to measure the taxonomy composition and abundances of APL patient’s sample [48]. Using the Venn Diagram package of R software (version 3.1.1), a Venn diagram was generated to show the shared and unique OTUs (the number of species detected) among groups [49]. Taxa abundances at different levels were statistically analyzed and plotted using QIIME. Furthermore, for keystone taxa screening, LEfSe (https://huttenhower.sph.harvard.edu/galaxy/ (accessed on 20 July 2021)) was used to detect differentially abundant genera [50]. A partial least squares discrimination analysis (PLS-DA) model was generated to identify key genera that were responsible for the differential distributions of gut microbiota between groups [51]. Mean value, standard deviation, and standard error (n = 3) were calculated using Excel 2010 (Microsoft Office 2010, Microsoft, Redmond, WA, USA).

### 2.5. Bacteroides Fragilis Str. 3783N1-6 Culture and Arsenic Resistance Assays

*B. fragilis str.* 3783N1-6 was purchased from the American Type Culture Collection (Manassas, VA). Cells were initially grown on agar plates to revive bacterial strains. Tryptic soy and 5% rabbit blood (Fisher Scientific, Pittsburgh, PA, USA) were initially added in agar. Bacterial in agar plates were grown in an anaerobic jar with GasPak^TM^ for 36 h at 37 °C (Appendix A) [52].

For the arsenic resistance experiment, when *B. fragilis str. 3783N1-6* grow to OD A_600nm_ = 0.3, indicated concentrations of trivalent arsenic As(III) (Sigma-Aldrich, Milwaukee, WI, USA) were added to *B. fragilis str. 3783N1-6* cultures with shaking. *B. fragilis* was grown in degassed Schneider’s insect medium with 20% heat-inactivated fetal bovine serum (Sigma-Aldrich, St. Louis, MO, USA). An anaerobic jar with GasPak^TM^ was used to keep 24 h anaerobic growing condition. Cells were collected if growing to an A_600nm_ = 1.0 [52].

### 2.6. Quantitative Reverse Transcriptase-Polymerase Chain Reaction (qRT-PCR) Analyses

According to the manufacturer’s instructions, an AccuScript High Fidelity 1st strand cDNA synthesis kit (Agilent, Santa Clara, CA, USA) was used to carry out cDNA synthesis. The primers designed using IDT’s Primer Quest software were purchased from Integrated DNA Technologies (IDT) (Coralville, lA). Genomic DNA of *B. fragilis str. 3783N1-6* was used to detect the efficiency of primers, and only efficiency between 98% to 102% primers was used in this study (Appendix A). To normalize gene expression, the cDNA of house-keeping gene 16S rRNA was used as a control. 1X iQSYBR Green supermix (Bio-Rad Laboratories, Inc., Hercules, CA, USA) was added to the mix to carry out the qPCR reaction with 10 μL of primers and 2 ng of template cDNA, and thermal cycling conditions are: 95 °C for 3 min denaturation followed by 40 cycles of 95 °C for 15 s and 65 °C for 20 s. The 2^−ΔΔCT^ method [53] was estimated for relative target gene amplicons with 16S rRNA as control. Error bars were calculated from the mean ± SD in triplicate.

### 2.7. Determination of Total Arsenic Concentration

Urine samples were digested using common methods to prepare solutions for measurement by hydride generation and arsenic concentrations were quantified by optical emission spectrometry with inductively coupled plasma (HG-ICP-OES). Fecal pellet samples (200 mg) in cap covering tubes were digested in concentrated HNO3 (68–70%) for 2h at room temperature, then heated at 95 °C for 1 h. Digested samples were diluted in ultra-pure water to a final HNO_3_ concentration of 5%. Final dilutions were centrifuged for 10 min at the highest speed to remove particulates, and the supernatant was collected for analysis. Arsenic was quantified by a hydride generation-atomic fluorescence spectrometer (HG-AFS, Guangzhou, China).

## 3. Results

### 3.1. Research Sample Assessment

The species accumulation curve was analyzed to fully understand community species composition and forecast species richness in biodiversity and community surveys (Appendix A). The species accumulation curve, where the upward tread at the end of the curve tended to be flat, indicated that the sampling quantity is sufficient. The RTU rank curve displayed the species diversity in samples, and at the same time explained the good richness and uniformity of species (Appendix A). Reflected by the length of the horizontal axis of the curve, samples A1, A4, and A5 of group A showed the highest richness with the wider the curve, the richer the species composition of the sample. Samples A1, A4, and A5 of group A had the highest degree of richness as shown by the length of the curve’s horizontal axis; the richer the sample’s species composition, the wider the curve. By contrast, B6, B15, B16, and B72 of group B showed steep and short curves, indicating lower richness and lower uniformity of species. The uniformity of species in the sample is reflected by the shape of the vertical axis of the curve. The flatter the curve, the higher the uniformity of species composition in the sample. The gut microbe species composition before arsenic exposure (group A: sample A1, A2, A4, A5, A12) demonstrates the richer composition and the higher uniformity, in which sample curve A4 was the widest and flattest one. Principal component analysis (PCA) was performed to compare the gut microbiome profiles of the A and B groups (Appendix A). Group A and Group B were separated by a 95% confidence interval (Cl) as the threshold to identify potential outliers in all samples. pcal1(45.31%) was considered as the main factor (maybe the states of patients’ illness individually) leading to the separation of the two groups.

### 3.2. Gut Microbiome Composition Changes with ATO Treatment

From 16S rRNA sequencing data, Figure 1A shows the identified gut bacteria in groups assigned at the phylum level. Before receiving arsenic treatment (Group A), Firmicutes (42.79%) and Proteobacteria (23.65%) were predominant in the gut bacteria of APL patients, followed by *Bacteroidetes* (15.88%), Actinobacteria (14.68%) and Others (1.992%). After As_2_O_3_ treatment, the gut microbiome in APL patients changed at the phylum level, showing that Firmicutes (59.55%) and *Bacteroidetes* (24.63%) were predominant, followed by Proteobacteria (10.89%) and Actinobacteria (2.689%). After arsenic exposure, Firmicutes and *Bacteroidetes* phyla increased by 55% and 40%, respectively, while Actinobacteria and Proteobacteria phyla decreased by 82% and 54%. Our finding revealed that Firmicutes and *Bacteroidetes* were dominant bacterial members after treatment in human gut microbial communities is consistent with prior direct observation studies on arsenic-treated mammals that have been previously published [35]. To gain insight into dominant taxa, a taxonomy tree visualized by GraPhlAn was applied and the most abundant taxa signified by the letters in the tree include Actinobacteria, *Bacteroidetes*, Firmicutes, Proteobacteria, and Verrucomicrobia, as shown in Appendix A.

The distribution of phyla between experimental individual samples was shown in Figure 1B with each column representing an individual APL patient sample. Individual bacterial phyla composition after arsenic treatment (B2, B4, B5, B6, B7, B8, B9, B10, B11, B12, B14, B15, B16, B17, B72) shifted to lower diversity and uniformity (evenness) compared with individuals before taking arsenic as treatment (A1, A2, A4, A5, A12). Similarly, at the class level (Figure 2A), individual fecal bacterial composition showed higher diversity and uniformity before taking arsenic as medicine compared with fecal samples after taking arsenic as medical treatment. Corresponding to the groups distribution of gut microbiome at the phylum level, the gut microbiome was overwhelmingly dominated by Firmicutes and *Bacteroidetes* after arsenic treatment. Most APL patients after arsenic treatment showed notably high Firmicutes composition, such as patient samples B6, B72, B9, B2, B7 and B16, excepting individual patient differences such as B5 (Figure 1B). Our observations and assignments of gut bacteria at phylum level after arsenic exposure are consistent with Dr. Lu’s previous reports that the gut microbiome was changed to the higher composition of *Bacteroidetes* [35].

By comparison with the database, SPECIES classification was also carried out for OTU abundance of groups at class and family levels (Figure 2A,B). The result clearly demonstrated that ATO treatment (B group) notably disturbed the evenness of gut microbial flora at the class level and family level corresponding to the phyla level. There was a higher abundance of Bacilli shifting from 17.2% to 47.9% after ATO treatment at the class level. Among upregulated classes, *Bacteroidia* increased from 19.3% to 25.5%. However, the class of Gammaproteobacteria decreased from 23.6% to 10.2%, and the class of Clostridia decreased from 24.5% to 11.5%. Likewise, there were serious uniformity disturbances at the family level showing that the family of *Enterococcaceae* increased from 4.2% to 43.9% and the family of *Bacteroidaceae* increased from 8.2% to 19.1% accompanied by the family of Enterobacteriaceae, Bifidobacteriaceae and Lachnospiraceae downregulation.

Meanwhile, at the family level (Figure 2B), we found that *Enterococcaceae* was dramatically increased from 4.2% to 44%, and *Bacteroidaceae* was upregulated 137% (0.19-0.08/0.08), but Bifidobacteriaceae and Enterobacteriacease were dramatically decreased. It suggested that the evenness of the gut microbiome of APL patients was decreased at class and family levels after AATO treatment.

The species abundance heat map is a graphical display of color gradient representing the relative abundance of species and clustering according to the abundance similarity of species or samples. Figure 2C demonstrated the differences in microbial communities before and after arsenic treatment. After arsenic treatment, the richness was increased in the family of Enterococaceae, *Bacteroidaceae*, Veillonellaceae, Verrucomicrobiaceae, Leuconostocaceae, Fusobacteriaceae, Paraprevotellaceae, Eubacteriaceae, Desulfovibrionaceae, etc. In contrast, gut microbiome richness was decreased after AATO treatment in the family of Enterobacteriaceae, Ruminococcaceae, Coriobacteriaceae, Streptococcaceae, Lachnospiraceae, Bifidobacteriaceae, *Lactobacillaceae*, etc. It is worth noting that the species *B. fragilis*, *Bacteroodes uniformis* and *Bacteroides ovatus* were upregulated (Figure 2D).

### 3.3. ATO Treatment Leads to Decrease Microbiome Diversity

For bacterial diversity analysis, the box chart of intergroup alpha diversity can more intuitively display the differences in intergroup alpha diversity. Wilcoxon test was used for comparison between the two groups (Figure 3). Appendix A shows the alpha diversity indices of Chao, Shannon, and Simpson. The Shannon and Simpson indices were significantly different before and after arsenic treatment (*p* = 0.019, *p* = 0.025 respectively), indicating lower diversity and evenness of gut microbiome after ATO treatment. The Chao richness index in the control group was higher than arsenic treated group, but there was no significant difference between groups by *t*-test. However, it indicated that the microbial community after arsenic treatment displayed a lower richness than that displayed in the control group although not significant (Wilcoxon test; *p* = 0.052 for observed richness).

### 3.4. Different Intestinal Microbe Analysis after ATO Treatment Predicts Keystone Taxa Microbe

By Wilcoxon rank-sum test and Kruskal-Wallis test, we identified that *Lactobacillaceae*, *Erysipelotrichaceae*, *Methanobacteriaceae*, *Peptostreptococcaceae* and *Clostridiaceae* were significantly changed at *p* < 0.05 (Figure 4). The taxonomic assignments and fold changes of gut bacterial components at family level that were significantly changed (*p* < 0.05) and listed in Appendix A with highlighted yellow color. In Figure 4, *Alcaligenaceae*, *Aerococcaceae*, *Enterococcaceae* and *Corynebacteriaceae* were downregulated before treatment when expressed by log2 A to B ratio (ratio <1, log2 ratio <0). By contrast, it indicated that *Alcaligenaceae*, *Aerococcaceae*, *Enterococcaceae* and *Corynebacteriaceae* were upregulated after treatment, with *Enterococcaceae* as the dominant family.

Irrespective of the abundance across space and time, microbial keystone taxa exert a considerable influence on microbiome structure and function. Table 2 displays the average relative abundance of each group and the significance of the difference test. At the class level, *Erysipelotrichi* and *Methanobacteria* were decreased after treatment with *p* = 0.011 and *p* = 0.015 respectively, which was consistent with the decrease in their family level, presented by *Erysipelotrichaceae* and *Methanobacteriaceae*, respectively (Figure 4). *Erysipelotrichales* keeps the same abundance from class to order level (Table 2), showing as *Erysipelotrichales* and *Erysipelotrichi* decrease to 0.13% after treatment. Similarly, *Methanobacteriales* (order level) and *Methanobacteria* (class level) all disappear after treatment. The genus Lactobacillus belongs to the phylum Firmicutes, class Bacilli, order II Lactobacillales, and family *Lactobacillaceae*. After ATO treatment, from family to genus to species level, *Lactobacillaceae* to Lactobacillus to *Lactobacillus mucosae* kept decreasing but were predicted to be keystone with *p* = 0.0083, *p* = 0.0083 and *p* = 0.023, respectively. *Bifidobacterium adolescentis* interpreted as a keystone with the lowest *p*-value 0.0043. Interestingly, we found *Bacteroidetes* phylum kept upregulated from *Bacteroidia* class to *Bacteroidales* order to *Bacteroidaceae* family to Bacteroides genus to *B. fragilis* species significantly (*p* < 0.001). Therefore, we evaluated *B. fragils* as keystone taxa as well as functional species.

### 3.5. Arsenic Resistance Genes Expression in Bacteroides Fragilis

Human intestinal microflora is a complex ecosystem colonized by ~500–1000 microbial species with the function to maintain immune and metabolic homeostasis and against pathogens [54]. Considering that Bacteroides spp. are among the predominant members of the human intestinal microflora (Figure 1, Table 2), the response of *Bacteroidetes* to arsenic exposure was investigated in this study. *B. fragilis* is an anaerobic bacterium that is a common component of human colon bacteria [55]. Arsenic resistance assay indicated *B. fragilis* 3783N1-6 is highly resistant to As(III), showing concentration-dependent (from 0 ppb to 100ppm) inhibition of cell growth (Figure 5A). Whole-genome sequence analysis indicated *B. fragilis* 3783N1-6 has a putative *ars* operon with eight genes (Figure 5B), including arsenic-related genes, *arsR*, *arsD*, *arsA*, *arsC*, and *arsB.* To examine whether these *ars* genes are involved in arsenic detoxification, the expression of each *ars* gene induced by As(III) was determined by qPCR (Figure 5B). It clearly showed all the *ars* genes expressions were upregulated with As(III) addition. At 10 ppm (0.13 mM) As (III) induced about 30-fold relative to 16s rRNA expression (to respond to the EPA’s usage, the arsenic concentrations in Figure 5B are expressed as ppm). The *arsA*, *arsB(acr3)* and *arsC* genes were expressed at nearly the same levels as each other, but the *arsD* genes were consistently expressed at about two-fold higher levels than the others. These results suggested that arsenic-responsive genes that confer resistance to inorganic arsenic may be responsible for Bacteroides keeping a high abundance in gut microbiome bacterium after arsenic treatment. These findings contribute to a better understanding of the mechanisms involved in arsenic metabolism in the gut microbiome and provide a strategy for the assessment of medical efficiency and adverse effects in APL.

### 3.6. Gut Microbiome OTU Numbers Were Associated with Arsenic in the Feces

We performed ICP-OES (JMT-H-060) and HG-AFS-2202E (JMT-H-057) analyses on arsenic concentrations in urine and feces. As sample limiting, we randomly selected a few urine and feces samples of APL patients to measure arsenic concentrations. After more than 7 days arsenic-containing medicine treatment, arsenic concentration in urine (B1, B4, B5, B6, B8, B9) was around 0.3–0.8 mg/L (average = 0.6 mg/L) and arsenic concentration in feces (B2, B4, B7, B12, B17) was around 0.02-1.03mg/kg (average = 0.33 mg/kg) (Appendix A). Arsenic concentration in the feces of APL patients after ATO treatment was associated with OTU numbers of the microbiome by Spearman and Kendall analysis with *p* < 0.05 (Figure 6).

## 4. Discussion

The toxicity of arsenic to non-target organs is the major drawback of the use of arsenic in medicine [10]. Therefore, there is a need to reveal arsenic adverse effects on the human body, including the microbiome. The use of ATRA and As_2_O_3_ led to a dramatic improvement in APL clinical outcomes [56]. This paper describes the disturbance of the gut microbiome considering AATO for the treatment of APL. Our result demonstrated that fecal microbial communities are distinct in patients of APL before and after As_2_O_3_ treatment. Following ATO therapy, gut microbiomes were found to be overwhelmingly dominated by Firmicutes and *Bacteroidetes* (Figure 1). The fecal microbiota composition in patients after taking medicines containing arsenic showed lower diversity and uniformity (Figure 2 and Figure 3). After treatment, *Bacteroides* were consistently upregulated (Table 2). In the most common gut microbiome *B. fragilis*, arsenic resistance genes were significantly induced by arsenic exposure in anaerobic pure culture experiments. We measured arsenic in the feces of APL patients with more than 7 days of treatment and arsenic concentration in feces was associated with OTU number of APL patients.

**Arsenic interacts with gut microbes.** Arsenic toxicity includes deleterious effects on gut microbiota, gastrointestinal disorders, immunological disturbances, disrupting metabolism, and compromising the host health, among which arsenic-induced perturbed gut microbiome communities that trigger systemic responses in diverse organs [57]. 16S rRNA gene sequencing revealed that arsenic exposure altered the composition of the intestinal microbiota from phylum to species, significantly affecting the Beta diversity of intestinal flora [58]. Corresponding to our research data, the phylum level of Firmicutes and *Bacteroidetes* shifted significantly after arsenic exposure in mouse models [58]. Bacteroides keep a high abundance in gut microbiome bacterium after arsenic treatment in our research. Similarly, there are research found that bacteroides remained relatively unchanged [35] or increased [33] after arsenic exposure in mouse models. There was one research carried out in Nepal, which has suffered from arsenic contamination for many years, to show that arsenic shaped the gut microbiome through its enrichment of arsenic volatilizing and pathogenic bacteria and depletion of gut commensals [34]. *B. fragilis* is characterized as one of the opportunistic pathogens and upregulated in our research in the gut of APL patients after ATO treatment. Arsenic may pose a broader human health risk than was previously known. Alternatively, the gut microbiome can participate in arsenic metabolism. Bacteria in the human gut can metabolize arsenic and influence the arsenical oxidation state, methylation status, thiolation status, bioavailability, and excretion [59]. In a mouse model, researchers find that a normal gut microbiome withstands arsenic stress by maintaining a healthier fecal microbial network and healthier fecal metabolome including amino acids, short-chain fatty acids, organic acids, and bile acid [36]. Our pure culture experiment support that arsenic-responsive genes may be responsible for Bacteroides keeping a high abundance in gut microbiome bacterium after arsenic treatment. The mouse model supported that long-term exposure to high As concentrations changed the expression of As resistance-related genes in intestinal microbes [58]. These discoveries indicate the potential of the gut microbiome for bio-detoxification of chronic arsenic exposure. ATO was also used to treat rheumatoid arthritis and ATO treatment modulated gut microbiota disorder and improved fecal metabolite abnormalities in a mouse model of rheumatoid arthritis [60]. Human bodies are variable in diet and living environment. It is unknown if the effects of arsenic on the human gut microbiome varied depending on inter-individual variance. How the complexities of the human gut bacterial community change while shifting before and after arsenic drug treatment is far from being fully understood.

**How did iv administration of arsenic influence the gut microbiome?** Arsenic is metabolized in the liver and bile and urine are the main routes for arsenic excretion [61]. At the hospital, APL patients were assigned to receive ATO and ATRA by intravenous (iv) until complete remission [16]. Arsenic through iv taking in rat model is metabolized in liver and excreted into bile by enterohepatic circulation [17], by which arsenic will end up in the gut. The route, dose, and chemical forms of arsenical administration will affect arsenic excretion and speciation from bile and urine: the metabolic speciation studies revealed that when exposed orally, arsenic was excreted into bile in rats, and methylarsenic-diglutathione (MADG) and/or dimethylarsenic acid (DMA(V)) were the main forms of arsenic species, summarized as iAs(III)- or iAs(V)-po (orally) rats; but when exposed intravenously, MADG and arsenic-triglutathione (ATG) are the main forms of arsenic species excreted into bile, summarized as iAs(III)- and iAs(V)-iv (intravenously) rats [17]. It was reported that after oral and iv administration, arsenic levels in tissues were similar following iv vs. oral administration, except lower in the intestine [62]. A physiologically based pharmacokinetic hamster and rabbits’ model for arsenic exposure including oral intake, iv injection, and intratracheal instillation was built. The tissue concentrations and the urinary and fecal excretions of the arsenic metabolites can be measured [63]. Another research demonstrated that arsenic exposure by intravenous caused the immediate appearance of arsenic in most tissues compared to gastrointestinal, and skin exposure pathways of arsenic in doses of 0.1 to 4.0 mg/kg, and intravenous administration showed a slower decrease of arsenic concentrations in time [64]. Early studies showed that iv or intraperitoneal injection in early gestation caused murine malformation when given a high dose of inorganic arsenic [65]. In our research, APL patients were treated with As_2_O_3_ by iv following 0.15 mg/kg daily arsenic trioxide. In that case, the patient’s daily intake of arsenic is around 7.5–9 mg based on an average body weight of 50–60 kg. Arsenic concentration in urine (B1, B4, B5, B6, B8, B9) was around 0.3–0.8mg/L (average = 0.6 mg/L) and arsenic concentration in feces (B2, B4, B7, B12, B17) was around 0.02-1.03mg/kg (average = 0.33 mg/kg) after 5 days of AATO treatment. We can calculate that arsenic excretion from urine was 0.9 mg/day based on average urine of 1500 mL, and arsenic excretion from feces was around 0.099 mg based on 0.3 kg feces as the average excretion amount.

**The effects of ATRA on the gut microbiome of APL patients cannot be ignored.** For newly diagnosed APL patients, As_2_O_3_ with or without ATRA was a highly effective treatment [66]. There are a few studies about the effects of ATRA on the human gut microbiome. One research reported that ATRA, as a key mediator of intestinal immunity [67], modulated the gut microbiome by maintaining tolerance to the intestinal microbiome [68]. Comparing pre- and post-AATO treatment, retinal metabolism and drug metabolism are predicted as functional difference pathways (Appendix A). In addition, xenobiotics biodegradation and metabolism, membrane transport, and lipid metabolism were highlighted in the Kyoto Encyclopedia of Genes and Genomes (KEGG) pathway (Appendix A). We predicted that arsenic contributed to gut microbiome disturbance even though arsenic was not the only medicine for APL patients. Our conclusion clarifies that arsenic-containing medicine disturbed human gut microbial flora to lower diversity and uniformity. However, since some microbial flora has been up-regulated, we predicted that those bacteria have arsenic detoxification mechanisms. In up-regulated bacteria *B. fragilis str.* 3783N1-6, the expression of arsenic resistance genes was all significantly increased which supports our hypothesis. Combined (ATO and ATRA) and single (only ATO or ATRA) drug exposure experiments can give more clues to figuring out the cause of microbiome change.

**Keystone taxa of human gut microbiome**, such as *B. thetaiotaomicron* [29], *B. fragilis* [27], *Helicobacter pylori* [69] and *Ruminococcus bromii* [70], exert considerable control on microbiome structure and functioning. In this research, arsenic resistance gene expression analysis for *B. fragilis* provides mechanistic evidence for the disturbance of the microbiome. Actually, it is hard to carry out experiments for keystone characterization. Comparing the effects of removal and/or addition of a potential keystone while removal and/or addition of other community members is the classic experiment to identify a keystone. In 1966, an ecologist, Robert T. Paine, first carried out a classic experiment for the tenet of keystone taxa at Makah Bay, Washington, USA: the researcher removed sea stars of a community and found that the removal had a dramatic impact on the shoreline ecosystem community and local biodiversity at Makah Bay [26]. In this research, based on 16S rRNA sequenced, the top 10 species were selected to show the average relative abundance of each group and the significance of the difference test by software. The question may be as follows: can we predict keystones? Researchers declaimed that better network inference tools and more validation experiments are needed before hub taxa in inferred networks can be classified as keystones [71]. We confirmed that *B. fragilis* can express arsenic resistance genes with As(III) exposure. However, it would be better to say *B. fragilis* is a functional species but not a keystone, which may need removal and/or additional experiments in a community. In science, experimental evidence often comes years after theoretical propositions. In addition to *B. fragilis str.* 3783N1-6, we searched the *ars* operon of other three representative human gut Bacteroides, *B. vulgatus* ATCC 8482, *B. fragilis* 3_1_12, and *B. thetaiotaomicron* VPI-5482 from NCBI databank. All of them were carrying arsenic resistance genes (Appendix A).

Lactobacillus and *Bifidobacteria* were considered probiotics for maintaining intestinal epithelial homeostasis and promoting health [72]. Our research demonstrated that *B. adolescentis* and *L. mucosae* were decreased after AATO treatment (*p* < 0.05). Significance analysis of different species in Table 2 demonstrated that *L. mucosae* were significantly (*p* = 0.023) decreased after AATO treatment. *L. mucosae* that belong to the phylum Firmicutes, class Bacilli, order *Lactobacillales*, and family *Lactobacillaceae* have been found on the mucosal surfaces of pigs, cows, and humans [73]. Currently, because of the metabolic capabilities and ability to colonize host mucosal niches, *L. mucosae* is considered a probiotic [74]. In addition, it was also able to modulate the immunological response and bile pool composition of the host [75]. Mucin-adhesion abilities of *L. mucosae* LM1in vitro are the reason that research characterized it as the probiotics, too [76]. LEfSe clustering tree also indicated that the *Lactobacillacea* family was a potential keystone taxa for AATO treatment (Appendix A). *Ars* operons were not found in the genome of *L. mucosae* LM1 (complete genome from NCBI databank https://www.ncbi.nlm.nih.gov/nuccore/CP011013.1, accessed on 20 March 2022). The abundance of *L. mucosae* was decreased after arsenic treatment may be due to its sensitive to As(III).

**AB group and CD group keep highly consistent in PLS-DA and key different species analysis.** Unlike animal exposure studies in which animals were maintained unified growth conditions and food supply, human bodies are variable in diet and living environment. It is unknown if the effects of arsenic on the human gut microbiome varied depending on inter-individual variance. Considering A and B groups did not have the same sample numbers and part samples before and after treatment were not from the same patient, we evaluated C and D groups in which four patients supplied their feces samples keeping before and after treatment (Table 1). Feces samples of the same patients before and after treatment were analyzed by 16S rRNA sequence. Unweighted-Unifrac cluster distance matrix for UPGMA clustering analysis was carried out to assess the similarity of samples (Appendix A). For samples under the same branch, the shorter the branch length between samples, the more similar the two samples were, and the higher the similarity of species composition was. The more distant the samples, the greater the difference in species composition. UPGMA clustering demonstrated that the feces sample of the same patients kept closest in Figure 5, which confirmed the precision of our sample-collecting.

Although samples of the A and B groups showed highly inter-individual differences (Figure 1B), A, B group and C, D groups were highly consistent in PLS-DA analysis (Appendix A). PLS-DS cluster of A groups included samples of A1, A12, A2, A4, and A5 (Table 1); B groups included samples of B2, B4, B5, B6, B7, B8, B9, B10, B11, B12, B14, B15, B16, B17, and B72; C groups included samples of A2, A4, A5, and A12; D groups included samples of B2, B4, B5, and B12. In addition, at the class level, *Erysipelotrichi* was the key different species for both the AB group and CD groups after treatment (Appendix A). Actinobacteria, *Coriobacteriia*, *Erysipelotrichi* and *Methanobacteria* were decreased after treatment, but Bacilli was upregulated, which was consistent in AB and CD groups. Due to the seriousness of APL cancer, APL patients will receive AATO treatment if they are hospitalized, it is difficult to gather feces samples from patients before starting AATO treatment. In the AB group, there are five samples before treatment and fifteen samples after treatment. To confirm the reliable and scientific results of AB group analysis, we analyzed the CD group samples that kept the same sample numbers before and after treatment. More importantly, to rigorously normalize samples pre- and post-treatment, A2 and B2, A4 and B4, A5 and B5, and A12 and B12 samples were from the same patient.

Correlation does not imply causation. The causal link between the number of OTUs in the gut microbiome and arsenic levels in the feces is difficult to assess. Although we measured arsenic in the feces as well as urine of APL patients, more research is needed to explore the causality relationship. We can’t exclude ATRA remark effects on human gut microbiome as APL patients are treated with arsenic plus ATRA. From KEGG pathway, retinal metabolism is predicted as functional difference pathways pre- and post-treatment (Appendix A). We characterized *B. fragilis* as a functional bacterial after AATO treatment, but it is challenging to quantify all functional bacteria. The reasons could be the complex nature of the gut microbiome, and the myriad of potential interactions of gut microbiome with human cells/tissue, such as selective pressure from the host and microbial competitors’ effects on microbes composition in the human gut [55]. In addition, for recruiting volunteer patients, the inclusion condition should be: APL patient who has not taken arsenic for treatment before being in the hospital, who agrees to take part in this research; the exclusion criteria were as follows: pregnancy or breastfeeding, suffering from metabolic or gastrointestinal diseases, and long-term use of antibiotics within the 2 months before sampling. All those criteria lead to only recruiting 22 volunteers. APL patients will receive AATO treatment as long as being hospitalized, so it is difficult to gather feces samples from patients before starting AATO treatment.

## 5. Conclusions

The role of microbial communities in human health functioning is unequivocal. As_2_O_3_ and ATRA were the first-line treatment for APL. Our research is not intended to be a comprehensive resource of current microbiome analysis tools, but provides a novel insight to improve our understanding of arsenic effects on human intestinal microbial flora following intravenous exposure for APL disease treatment. The human body and intestinal microbes have long been mutually beneficial. The influence of xenobiotics on the human gut microbiome was required for the assessment of potential impacts on human health. Ethics, animal welfare theory, and toxicity testing in the twenty-first century all need to be considered. In toxicology, it is hard to investigate human arsenic exposure effects on the gut microbiome. Meanwhile, time-consuming, bigger sample size and variable exposure do make epidemiological research harder to advance compared to clinical active exposure. Our research accomplishes two goals at once: taking advance of clinical treatment to exploit arsenic exposure effects on the human gut microbiome and requiring APL patients to take arsenic as an effective medication, preventing passive exposure to our research samples. What’s more, human waste samples used for screening include urine and feces, which are not only simple to collect but also painless for the patient. This research provides evidence for the relationships between arsenic exposure and changes in the human microbiota, and even extends to arsenic-related health outcomes in APL. Functional species *B. fragilis* were considered as potential keystone taxa that may apply to be a biomarker for the gut microbiome disturbance of APL patients after AATO treatment.

## Figures and Tables

**Figure 1 toxics-11-00458-f001:**
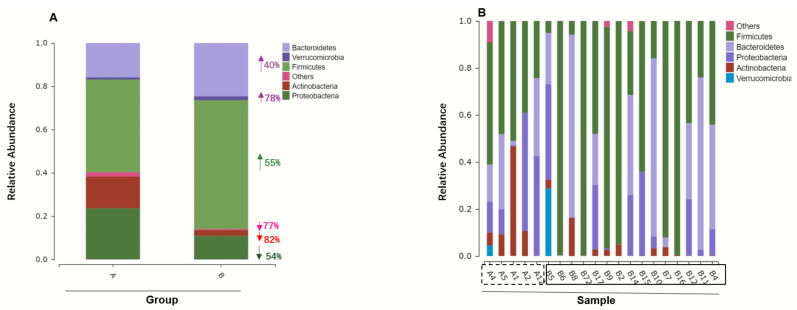
Histogram of microbial community composition pre- and post-arsenic treatment. (**A**) Distribution of phyla level between experimental groups. The arrows beside the bar indicate the relative changes in the abundance of the corresponding phylum. (**B**) Family level of gut microbiome composition pre- and post-treatment. Samples (from B6 to B5) in the solid box represent group B after arsenic treatment, samples (from A4 to A12) in the dashed box represent group A before arsenic treatment.

**Figure 2 toxics-11-00458-f002:**
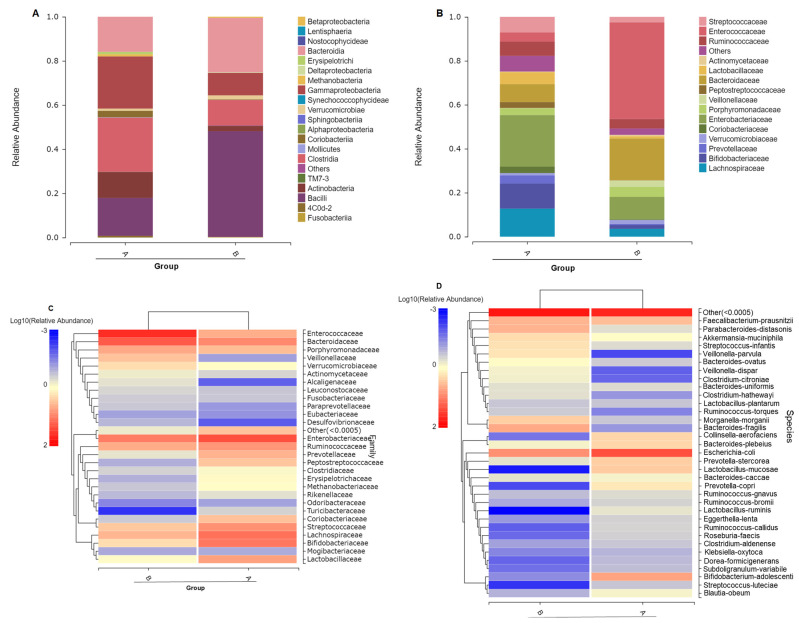
The gut microbial composition changes in experimental groups before and after treatment. Distribution of class (**A**) and family (**B**) level between experimental groups (the histogram of all species were drawn at the phylum level. Starting from the class level, species whose species abundance was lower than 0.5% in all samples and those not annotated in this classification level were merged into Others). The species abundance heat map at family (**C**) and species level (**D**) of the top 35 microbiotas.

**Figure 3 toxics-11-00458-f003:**
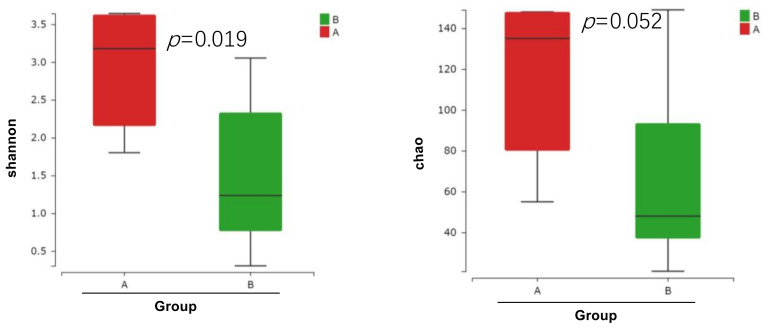
The gut microbial changes in experimental groups after treatment showed by the boxplots of alpha diversity differences in groups.

**Figure 4 toxics-11-00458-f004:**
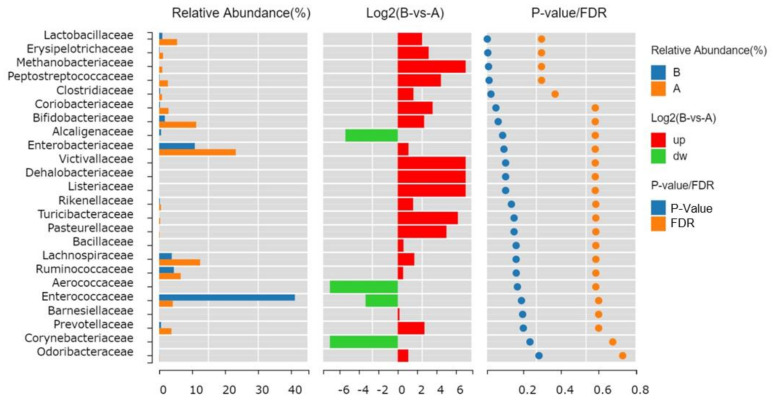
Species difference analysis at a family level between groups. In the middle is the log2 (A group: B group) value of the average relative abundance ratio of the same species in the two groups; The figure on the right shows *p*-value and FDR values obtained by the Wilcoxon test.

**Figure 5 toxics-11-00458-f005:**
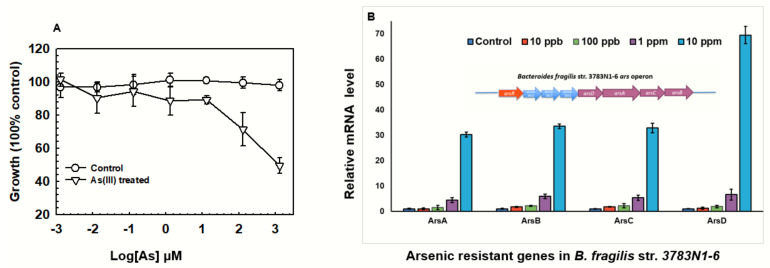
*B. fragilis str.* 3783N1-6 culture and dose-dependent induction of *ars* mRNAs. (**A**). Growth curve of *B. fragilis str.* 3783N1-6. AS_2_O3 exposure concentrations were 1 ppb, 10 ppb, 100 ppb, 1 ppm, 10 ppm and 100 ppm converted to a log value of molarity unit. Cultural conditions can be found in material and method parts. Data are the mean ± SE (n = 3). (**B**). Transcriptional analysis of the *arsA*, *arsB*, *arsC*, and *arsD* genes induced by As(III). *B. fragilis str.* 3783N1-6 were treated with increasing concentrations of sodium arsenite [As(III]). Total RNA isolated and qPCR was performed with *ars* gene-specific primers. Experiments were repeated three times. Arrows represent *ars* operon of *B. fragilis str.* 3783N1-6 confirmed in NCBI databank https://www.ncbi.nlm.nih.gov/nuccore/596269698 (accessed on 20 March 2022).

**Figure 6 toxics-11-00458-f006:**
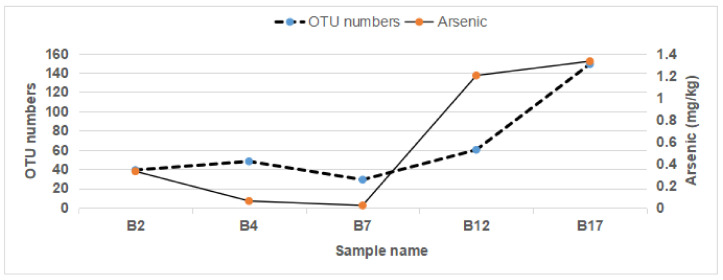
Curves of Arsenic concentration in feces and gut bacterial OTU numbers. Arsenic concentration in feces of B2, B4, B7, B12 and B17 were measured by AFS-2202E (JMT-H-057) analyses. As well, OTU numbers were calculated by QIIME2-DADA2 (Divisive Amplicon Denoising Algorithm).

**Table 1 toxics-11-00458-t001:** Feces samples of APL patients before and after arsenic treatment.

	A: Before Arsenic Treatment	C: The Same Patient before Arsenic Treatment	
	B: After Arsenic Treatment	D: The Same Patient after Arsenic Treatment	
Sample Name	Group 1	Group 2	Treatment Results
A1	A		no second visit
A12	A	C	CR
A2	A	C	CR
A4	A	C	CR
A5	A	C	CR
B10	B		CR
B11	B		CR (+7 chromosome)
B12	B	D	CR
B14	B		CR
B15	B		CR
B16	B		CR
B17	B		no second visit
B2	B	D	CR
B4	B	D	CR
B5	B	D	CR
B6	B		CR
B7	B		CR
B72	B		CR
B8	B		DEAD (turn to AML)
B9	B		CR
B1			DNA con too low
A3			

+7 chromosome represents that there is one more 7th chromosome in the patient. AML represents another kind of leukemia-Acute myeloid leukemia (AML).

**Table 2 toxics-11-00458-t002:** Key and functional species analysis in *Bacteroidetes*. *Bacteroidetes* from phylum to species were selected to display the significant differences between A and B groups (*p*-value < 0.001, *** is marked; 0.001 ≤ *p*-value ≤ 0.01, ** is marked; 0.01 < *p*-value ≤ 0.05, * is marked; If *p*-value > 0.05, it is not marked). Software used: R(V3.4.1).

Levels	Keystone/Functional Species	A (%)	B (%)
Phylum	Euryarchaeota (*p* = 0.015) *Bacteroidetes* ***	0.00% 16.52%	0.94% 26.27% up
Class	*Erysipelotrichi* (*p* = 0.011) * *Methanobacteria* (*p* = 0.015) * *Bacteroidia* ***	1.16% 0.94% 16.52%	0.13% 0.00% 26.27% up
Order	*Erysipelotrichales* (*p* = 0.011) * *Methanobacteriales* (*p* = 0.015) * *Bacteroidales* ***	1.16% 0.94% 16.52%	0.13% 0.00% 26.27% up
Family	Lactobacillacea (*p* = 0.0083) ** *Bacteroidaceae* ***	5.4% 8.92%	0.95% 20.38% up
Genus	Lactobacillus (*p* = 0.0083) ** *Bacteroides* ***	3.6% 8.92%	0.94% 20.38% up
Species	*Bifidobacterium adolescentis* (*p* = 0.0043) ** *Lactobacillus mucosae* (*p* = 0.023) * *Bacteroides fragilis* (*p* = 0.007) **	4.9% 2.6% 0.06%	0.04% 0.002% 4.83% up

## Data Availability

Not applicable.

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
