# Peer review of "Arsenic-Containing Medicine Treatment Disturbed the Human Intestinal Microbial Flora"

_toxics, 2023, doi:10.3390/toxics11050458_

Round 1

Reviewer 1 Report

Review on the manuscript of Li, J. et al.,: “Arsenic-containing Medicine Treatment Disturbed the Human Intestinal Microbial Flora”.

This manuscript explores how arsenic exposure in acute promyelocytic leukemia patients (APL) influence the gut microbiome. Authors show that fecal microbial communities are distinct in APL patients before and after As2O3 treatment. Particularly, Bacteroides were consistently upregulated after treatment of APL patients with As2O3. In addition, in vitro experiments with the most common gut bacteria, B. fragilis, showed an induction of arsenic resistance genes following arsenic exposure in anaerobic pure cultures.

The data shown in the manuscript is clear. However, some issues arise to me in its current format. So, I hope the authors find the following comments and suggestions useful.

1 - For this study, only APL patients were used. It would be of great interest to have included in the study control, healthy individuals to compare the gut microbiome with that of APL patients before starting arsenic treatment. Do the authors have an explanation for the no inclusion of healthy control subjects in the study.

2 – I assume the data shown in figure 1A and figure 2A-B came from individuals that were tested before and after arsenic treatment and from individuals that were tested only before or only after arsenic treatment. Therefore, how would look like the data only from individuals that were tested before and after arsenic exposure (the 4 individuals “C”)?

3 - In figure 1A and 1B, I recommend the authors to use the same color for each microbial community (for example, in figure 1A Bacteroidetes are in green, while in figure 1B green corresponds to Firmicutes).

4 – In the results section, authors say “Arsenic resistance assay indicated B. fragilis 3783N1-6 is highly resistant to As(III), showing as concentration-dependent (from 0 ppb to 100ppm) growth (Fig. 5A)”. Maybe authors should clarify this sentence to “Arsenic resistance assay indicated B. fragilis 3783N1-6 is highly resistant to As(III), showing a concentration-dependent (from 0 ppb to 100ppm) inhibition of cell growth (Fig. 5A)”.

Reviewer 2 Report

toxics-2368952-peer-review-v1

-Abstract: L 14:  Find: Effects of arsenic-containing medicine exposure on gut microbiome is unknown

Replaced with:

Few investigations have been conducted to understand the effect of arsenic-containing medicine exposure on gut microbiome.

-Abstract: L 17:  Find: samples from APL patients

Replaced with: samples from acute promyelocytic leukemia

Abstract: L 21:  Find: showing by the alpha diversity indices of Chao, Shannon, and Simpson. Gut microbiome OTU

Replaced with: Gut microbiome operational taxonomic unit (OTU)

- Find: L 76: . Bacteroides fragilis (B. fragilis) is isolated from anaerobic infections

Replaced with: Bacteroides fragilis is isolated from anaerobic infections

- Find: L 81: Bacteroides thetaiotaomicron (B. thetaiotaomicron), an anaerobic symbiont

Replaced with: Bacteroides thetaiotaomicron (formerly Bacillus thetaiotaomicron), an anaerobic symbiont

-Find: L 97:  Find: ingested arsenicals interacted with gut microbiome of mice, causing Firmicutes bacteria to significantly decline while Bacteroidetes families remained relatively unchanged. Insert suitable reference (s).

 - Introduction Section:

The introduction provides sufficient background and recent references should be cited.

- Find: L 109 and 110: Even while studies using mice or a simulated gastro intestinal tract have shown how arsenic alters the gut microbiome (Yu et al. 2016). Insert another recent reference (s).

-Find: L 380: Lactobacillaceae to Lactobacillus to Lactobacillus mucosae (L. mucosae) kept

Replaced with: Lactobacillaceae to Lactobacillus to Lactobacillus mucosae kept

-Find: L 444 all-trans retinoic acid (ATRA)] for the treatment of APL.

Replaced with: ATRA] for the treatment of APL.

-Find: L530:  Bifidobacterium adolescentis

Replaced with: B. adolescentis

- Check the format of all references. Check reference Numbers 28, 40, and 41

-Insert recent (2023) suitable reference (s) to the Discussion Section

e.g.  

Chen L, Li C, Zhong X, Lai C, Zhang B, Luo Y, Guo H, Liang K, Fang J, Zhu X, Zhang J. The gut microbiome promotes arsenic metabolism and alleviates the metabolic disorder for their mammal host under arsenic exposure. Environment International. 2023 Jan 1;171:107660.

-          Improve the resolution of Figure 2.

-          Improve the title of Table 2.

-          Find: L392: Arsenic resistance genes expression in Bacteroides. fragilis

-          Replaced with: Arsenic resistance genes expression in Bacteroides fragilis

Discussion Section

-          Insert: Arsenic can alter the human gut microbiome by promoting the growth of arsenic-metabolizing bacteria like Desulfovibrio, Bilophila, and Bacillus (Brabec et al., 2020). Arsenic induces structural and compositional colonic microbiome change and promotes host nitrogen and amino acid metabolism (Dheer et al., 2015).

Brabec JL, Wright J, Ly T, Wong HT, McClimans CJ, Tokarev V, Lamendella R, Sherchand S, Shrestha D, Uprety S, Dangol B. Arsenic disturbs the gut microbiome of individuals in a disadvantaged community in Nepal. Heliyon. 2020 Jan 1;6(1):e03313.

Dheer R, Patterson J, Dudash M, Stachler EN, Bibby KJ, Stolz DB, Shiva S, Wang Z, Hazen SL, Barchowsky A, Stolz JF. Arsenic induces structural and compositional colonic microbiome change and promotes host nitrogen and amino acid metabolism. Toxicology and applied pharmacology. 2015 Dec 15;289(3):397-408.

-          Improve the discussion section e.g. L 454: The title "How did iv administration of arsenic influence gut microbiome? "

-          L 493: The results mentioned in this research are affected only by arsenic or by ATRA or by both are still unknown. Reconstruct the sentence

Minor editing of English language are required
